METHODS

# QoALa: A comprehensive workflow for viral quasispecies diversity comparison using long-read sequencing data

Nakarin Pamornchainavakul◉*, Declan C. Schroeder, Kimberly VanderWaal

Department of Veterinary Population Medicine, University of Minnesota, St. Paul, Minnesota, United States of America

* pamor001@umn.edu

## Abstract

The concept of viral quasispecies refers to a constantly mutating viral population occurring within hosts, which is essential for grasping the micro-evolutionary patterns of viruses. Despite its high error rate, long-read sequencing holds potential for advancing viral quasispecies research by resolving coverage limitations in next-generation sequencing. We introduce a refined workflow, QoALa, implemented in the *longreadvqs* R package. This workflow begins with nucleotide position-wise noise minimization of read alignments and sample size standardization, and extends to viral quasispecies comparison across related samples with integrated visualization capabilities. Benchmarking on simulated SARS-CoV-2 and HIV-1 datasets demonstrated that QoALa consistently outperformed existing error-correction methods in recovering quasispecies composition, particularly in preserving nucleotide diversity and hierarchical population structure. Real raw read samples from five studies of different viruses (HCV, HBV, HIV-1, SARS-CoV-2, and IAV), sequenced by major long-read platforms, were also used to evaluate these approaches. The comparative results provide novel insights into intra- and inter-host diversity dynamics in various scenarios and unveil rare haplotypes not reported in the original studies, underscoring the versatility and practicality of our methodology.

## Author summary

Viruses evolve rapidly because their genetic material accumulates mutations during replication. As a result, viruses within a single infected host often exist as a mixture of closely related genetic variants known as a viral quasispecies. This diversity plays an important role in how viruses adapt, spread between hosts, and evade immune responses. Understanding these complex viral populations requires sequencing technologies that can capture detailed genetic variation. Traditional short-read sequencing often cannot reconstruct complete viral variants, while long-read sequencing can capture full genes but introduces more

provided the original author and source are credited.

**Data availability statement:** All real raw read sequencing data used as example datasets for this study are publicly available at the National Center for Biotechnology Information's Sequence Read Archive (NCBI SRA), as well as being part of previously published publications. Accession numbers and corresponding publications were tabulated in S1 Table. Simulated data used for benchmarking (limited to SARS-CoV-2 S and HIV env) — including replicates of ground truth, uncorrected reads, and reads corrected by different methods — are available at https://github.com/NakarinP/longreadvqs/tree/7184e665ee51b4e434d3b7b-7c129d4d5745de1f7/sim_data. Code availability All codes used in this study are available as an R package longreadvqs and accessible via the CRAN repository at https://cran.r-project.org/web/packages/longreadvqs/index.html.

**Funding:** 4. This work was supported by the Ecology and Evolution of Infectious Disease program jointly funded by the U.S. Department of Agriculture National Institute of Food and Agriculture (NIFA; https://www.nifa.usda.gov/), the National Science Foundation (NSF; https://www.nsf.gov/), and the National Institutes of Health (NIH; https://www.nih.gov) under award number 2019-67015-29918 to N.P., K.V., and D.C.S. Additional support for this study was provided by the Intramural Research Program of the U.S. Department of Agriculture, National Institute of Food and Agriculture, Data Science for Food and Agricultural Systems Program (grant number 2023-67021-40018 to K.V.). The funders had no role in study design, data collection and analysis, decision to publish, or preparation of the manuscript.

**Competing interests:** The authors have declared that no competing interests exist.

sequencing errors that complicate analysis. In this study, we developed QoALa, a computational workflow implemented in the *longreadvqs* package, to help researchers analyze viral quasispecies from long-read sequencing data. Our approach reduces sequencing noise, standardizes sample sizes, and provides tools to compare viral populations across samples. Using both simulated data and real sequencing datasets from several viruses, we show that our workflow can more accurately recover viral population structure and reveal rare genetic variants. This approach provides a practical framework for studying viral evolution from long-read sequencing data and facilitates deeper investigation of how viral populations diversify and adapt during infection and transmission.

## Introduction

The emergence of severe acute respiratory syndrome coronavirus 2 (SARS-CoV-2) in 2019, alongside historical pandemics, emphasizes RNA viruses, with their characteristically high evolutionary rates, as the most concerning group of pathogens. Their remarkable adaptability stands as a primary factor behind their widespread success [1]. Their rapid mutation typically stems from the lack of proofreading activity during genome replication in most RNA viruses, a key driver of genetic variability [2]. Although coronaviruses encode a proofreading exonuclease that reduces replication errors, they still exhibit substantial mutation rates due to high replication frequency and population size [3]. This variability manifests as a cloud of closely related genetic variants of the virus within the viral population infecting a single individual, which is known as viral quasispecies [4]. Studies elucidating quasispecies dynamics in relation to virus adaptation have significantly enhanced our understanding of RNA virus microevolution, virus-host interactions, and the ability of viruses to adapt to evade host immunity [5–11].

The viral quasispecies concept has been referred to in nearly one thousand virology studies, notably for hepatitis C virus (HCV), hepatitis B virus (HBV), type 1 human immunodeficiency virus (HIV-1) [12], where some viral variants correlate with antiviral drug resistance [13–16]. The advent of deep sequencing technology, alongside improvements in *de novo* viral genome assembly techniques, has enhanced our ability to detect minority variants within the viral cloud that may hold clinical significance [17–20]. While next-generation sequencing (NGS) has been widely employed for viral quasispecies analysis [21,22], one of the persisting challenges is the reconstruction of continuous viral quasispecies haplotypes—sets of identical genomic sequences. This challenge arises due to the short length of NGS reads (typically 100 – 400 bases), which often do not cover the entire targeted gene or genome, necessitating short-read assembly techniques that are often unable to discern between haplotypes that are closely related [19,23–26].

Long-read sequencing technology provides a solution to the coverage limitations of NGS technology in viral quasispecies research [23], yielding read lengths typically exceeding 1,000 bp. Few analytical workflows or software exist that are tailored for

viral quasispecies exploration using long-read data, and these primarily focus on single nucleotide variant (SNV) calling, haplotype reconstruction, or quasispecies profiling on an individual sample basis [27–35]. However, these packages do not fully address the unique challenges or harness the full potential of long-read sequencing for viral quasispecies analysis. One challenge is the higher error rates (5–20%) compared to NGS (~1%) [36], affecting haplotype reconstruction accuracy. Efforts in post-sequencing error correction have spurred the development of numerous tools for improving sequencing reads generated by Oxford Nanopore Technologies (ONT) or Pacific Biosciences (PacBio) [36–40]. However, even with lower error rates, the rapid mutation rate of RNA viruses means that longer reads often differ by only one or two bases, and thus are called a unique haplotype. This ultimately results in a large number of haplotypes that each occur only once within the data, and obscures our ability to discern structure the viral quasispecies.

Sequencing read depth and length also significantly influence viral quasispecies diversity measures [23,41,42]. Certain diversity metrics, such as Shannon entropy and mutation frequency, are dependent on sample size or read depth [42,43]. As yet, it is uncertain how such diversity metrics are influenced by sampling depth, particularly when comparing diversity between samples with different depths or down-sampled to the same depth, or even what minimum depth is needed to attain a robust measure of diversity. In addition, the longer the gene or genome length used for quantifying quasispecies, the greater the number of unique haplotypes or singletons due to the increased detection of mutations along its length. This variability may introduce bias, particularly when comparing viral quasispecies profiles across longitudinal or related samples.

In response to these challenges, we developed *longreadvqs*, an R package designed for viral quasispecies comparison using long-read data that can be applied to any RNA virus. This package creates a customizable workflow for long-read noise-minimization and down-sampling as well as for grouping related haplotypes into operational taxonomic units (OTUs). In addition, this package can analyze multiple samples together, identifying common haplotypes or OTUs that recur in different samples. This feature is particularly useful when analyzing longitudinal samples collected from a single individual or multiple individuals that may be epidemiologically linked. Our proposed analytical workflow, QoALa (Quasi-species Optimized & Adaptive Long-read Analysis), was evaluated using simulated and publicly available ONT and PacBio long-read datasets. Benchmarking with simulated data, we compared QoALa's ability to recover quasispecies structure against existing correction tools. Using real experimental and clinical datasets, we assessed sensitivity of results to read downsampling and demonstrated QoALa's ability to reveal quasispecies-level insights.

## Results

### QoALa workflow and example datasets

The goals of the QoALa workflow are to standardize and comprehensively compare viral quasispecies and OTU profiles across multiple samples based on user customization. For each comparison, equal length read alignments from the same gene or genomic region of interest (Fig 1a) are separately imported from any genome assembly pipeline's output (see examples in Methods) and noise-minimized using position-wise nucleotide base replacement in the "*vqsassess*" function (Fig 1b). The noise-minimization step replaces potential erroneous nucleotides, defined as SNVs with a frequency lower than n% of the total read depth (cut-off percentage), with either the majority base or the dominant haplotype's base at that position. This cut-off percentage can be determined using prior information such as the sequencing error rate, or by observing the change in the percentage of singleton haplotypes at different percentage cut-offs (see Methods and Fig 1a). Additionally, when comparing multiple samples, all alignments must be randomly down-sampled by the "*vqsassess*" function to achieve equivalent read depth (the shallowest depth among all samples is recommended) to reduce bias in diversity measurement caused by sample size disparity. SNV profiles of the samples visualized by the "*snvcompare*" function should be inspected to evaluate the result of noise-minimization (Fig 1c). Finally, prepared samples are pooled by the "*vqscompare*" function to (1) identify common haplotypes, (2) reclassify haplotypes into OTUs, (3)

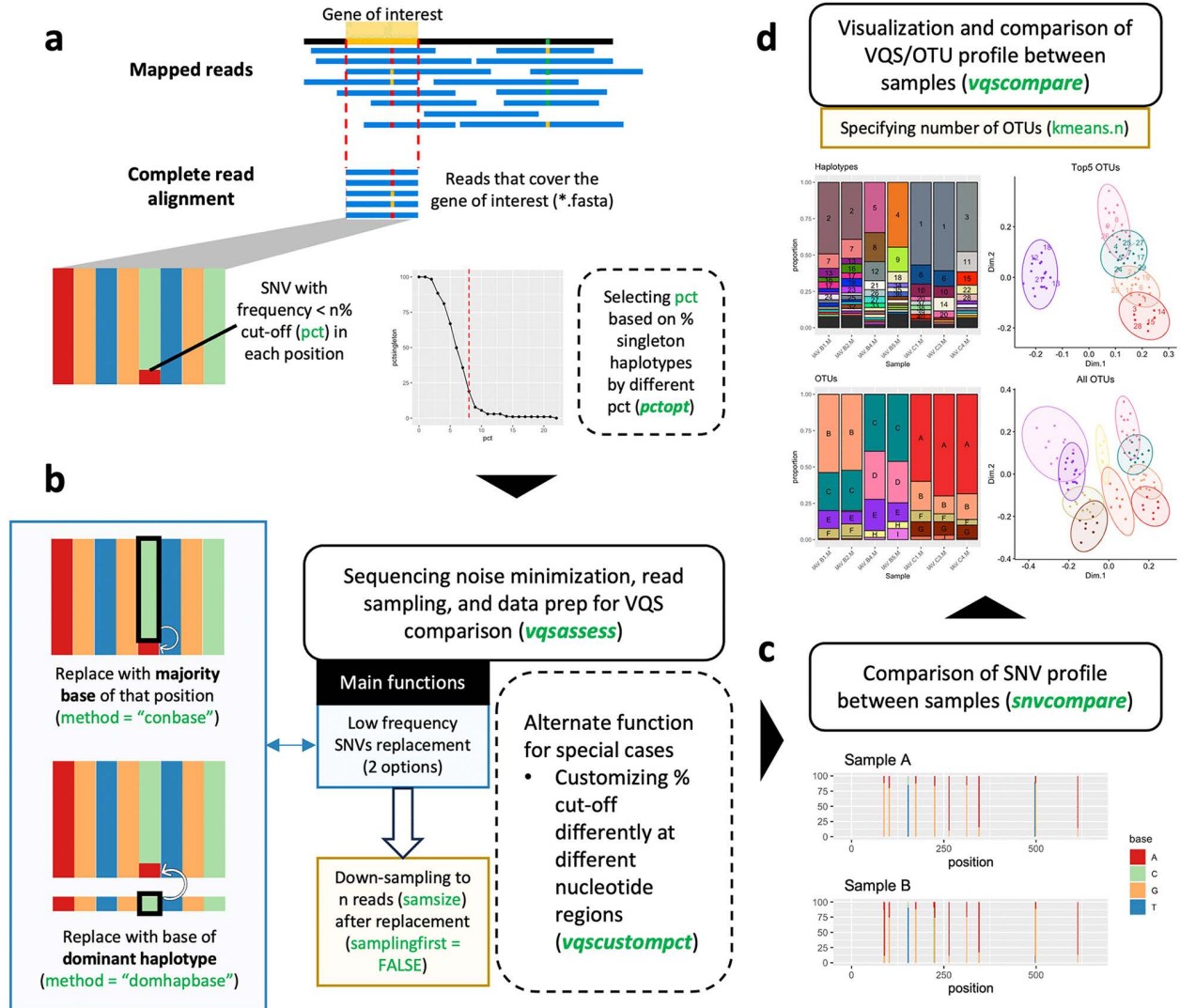

**Fig 1. QoALa workflow and main functions of *longreadvqs* package. a**, Sequencing read alignment that completely covers a gene of interest is used as input for subsequent analyses in the *longreadvqs* package. The percentage cut-off, which is used to define low-frequency SNVs potentially as noise, can be estimated from prior knowledge or the percentage of singleton haplotypes in the alignment. **b**, Sample preparation involves noise-minimization, where nucleotide bases with frequencies lower than the percentage cut-off are replaced with the majority base (recommended) or the base of the dominant haplotype at the respective position, followed by read down-sampling for sample size standardization. The percentage cut-off can also be customized for error-rich regions. **c**, The SNV profile after sample preparation should be compared between samples to visually check for remaining potential noise in the alignment. **d**, All prepared samples are pooled to identify shared haplotypes and reclassify them into OTUs based on genetic distance. Green texts indicate key functions (italicized) and arguments in the *longreadvqs* package. Solid boxes represent main steps in the QoALa workflow, while dashed boxes denote optional steps.

visualize diversity profiles and genetic relationships among samples ([Fig 1d]), and (4) summarize quantitative diversity metrics.

To benchmark the QoALa workflow under controlled conditions, we generated simulated long-read viral quasispecies datasets for SARS-CoV-2 and HIV-1 with known ground-truth haplotype composition. Simulations comprised five haplotypes with fixed frequency distributions across five independent replicates and sequences were generated *in silico* using

representative ONT and PacBio error models. These datasets enabled direct, quantitative comparison of QoALa with alternative correction strategies—including Strainline [33], Racon [44], and uncorrected reads—by assessing accuracy in recovering true haplotype and OTU structures across sequencing technologies.

We further evaluated the QoALa workflow using read alignments assembled from publicly available long-read datasets representing five of the most extensively studied viruses in the viral quasispecies field: HCV, HBV, HIV-1, SARS-CoV-2, and influenza A virus (IAV). These datasets spanned diverse sequencing technologies (PacBio and ONT), sampling designs (cross-sectional and short- or long-term longitudinal), and depths of coverage per gene of interest, ranging from 130 reads in the IAV M gene segment (S6 Table) to 854,147 reads in the SARS-CoV-2 ORF3a gene (S5 Table). In this setting, QoALa was used to assess sensitivity to read downsampling and to re-examine quasispecies structure at standardized depths, enabling direct comparison across samples. Estimated sequencing error levels, inferred from singleton haplotype proportions prior to noise minimization, varied widely—from ~25% in the SARS-CoV-2 ORF3a sample to nearly 100% in the HCV and HIV-1 samples (S1 Fig). This reanalysis revealed quasispecies-level diversity patterns that were not revealed in the original studies and underscores the need for customized long-read preprocessing, particularly with respect to cut-off percentage and sample size.

## QoALa improves recovery of viral quasispecies diversity and composition across sequencing platforms

Across simulated SARS-CoV-2 (*S* gene) and HIV-1 (*env* gene) quasispecies datasets, error-correction approaches showed metric-dependent performance patterns that were largely consistent across sequencing platforms (Fig 2c–2e). Based on Bray–Curtis dissimilarity, which captures agreement in both composition and relative abundance compared to the ground truth, QoALa achieved the lowest or near-lowest dissimilarity at the OTU level for both SARS-CoV-2 *S* and HIV-1 *env* datasets, generally reducing Bray–Curtis deviation by approximately 2–4-fold relative to uncorrected reads. For the most recent platforms (ONT 2023 and PacBio HiFi), QoALa and Strainline, and in some cases Racon, showed comparable OTU-level Bray–Curtis performance with no significant differences. However, all OTU analysis (regardless of which pipeline was used for error-correction) relied on the QoALa workflow, as OTU analysis is not a functionality of Strainline or Racon. At the haplotype level (i.e., no grouping of reads into OTUs), Bray–Curtis values were uniformly high and close to one for both viruses across all tools, indicating that none of the approaches reliably recovered true haplotype compositions (Fig 2c).

Evaluation using the Gini–Simpson index ($H_{GS}$), which summarizes within-sample diversity and evenness independent of quasispecies composition, also revealed contrasting trends between taxonomic resolutions. At the haplotype level, Strainline and Racon generally showed smaller deviations from ground-truth $H_{GS}$ than QoALa for both SARS-CoV-2 *S* and HIV-1 *env* datasets, reflecting better preservation of population evenness, though the magnitude and statistical significance of these differences varied by sequencing platform. In contrast, at the OTU level—where genetically related haplotypes are grouped—QoALa consistently yielded the smallest $H_{GS}$ deviations across both viruses and most sequencing platforms. While Strainline and Racon occasionally showed numerically smaller deviations than uncorrected reads at the OTU level, these differences sometimes was statistically indistinguishable from uncorrected reads (Fig 2d). These platform-dependent patterns indicate that the advantages of Strainline and Racon at the haplotype level do not consistently translate to OTU-level analyses.

Nucleotide diversity ($\pi_m$), a sequence-based measure of underlying genetic and functional heterogeneity, showed the clearest and most consistent separation among tools. Across both SARS-CoV-2 *S* and HIV-1 *env* datasets, QoALa exhibited significantly smaller deviations from true $\pi_m$ than all other approaches at both haplotype and OTU levels, improving $\pi_m$ recovery by two to three orders of magnitude relative to uncorrected reads, i.e., from ~$3 \times 10^{-2}$–$10^{-1}$ in uncorrected reads to ~$10^{-4}$–$10^{-3}$. While Strainline and Racon significantly reduced $\pi_m$ deviation at the haplotype level, their OTU-level $\pi_m$ deviations remained similar to uncorrected data (Fig 2e), indicating persistent loss of fine-scale genetic variation that was minimized by the QoALa workflow.

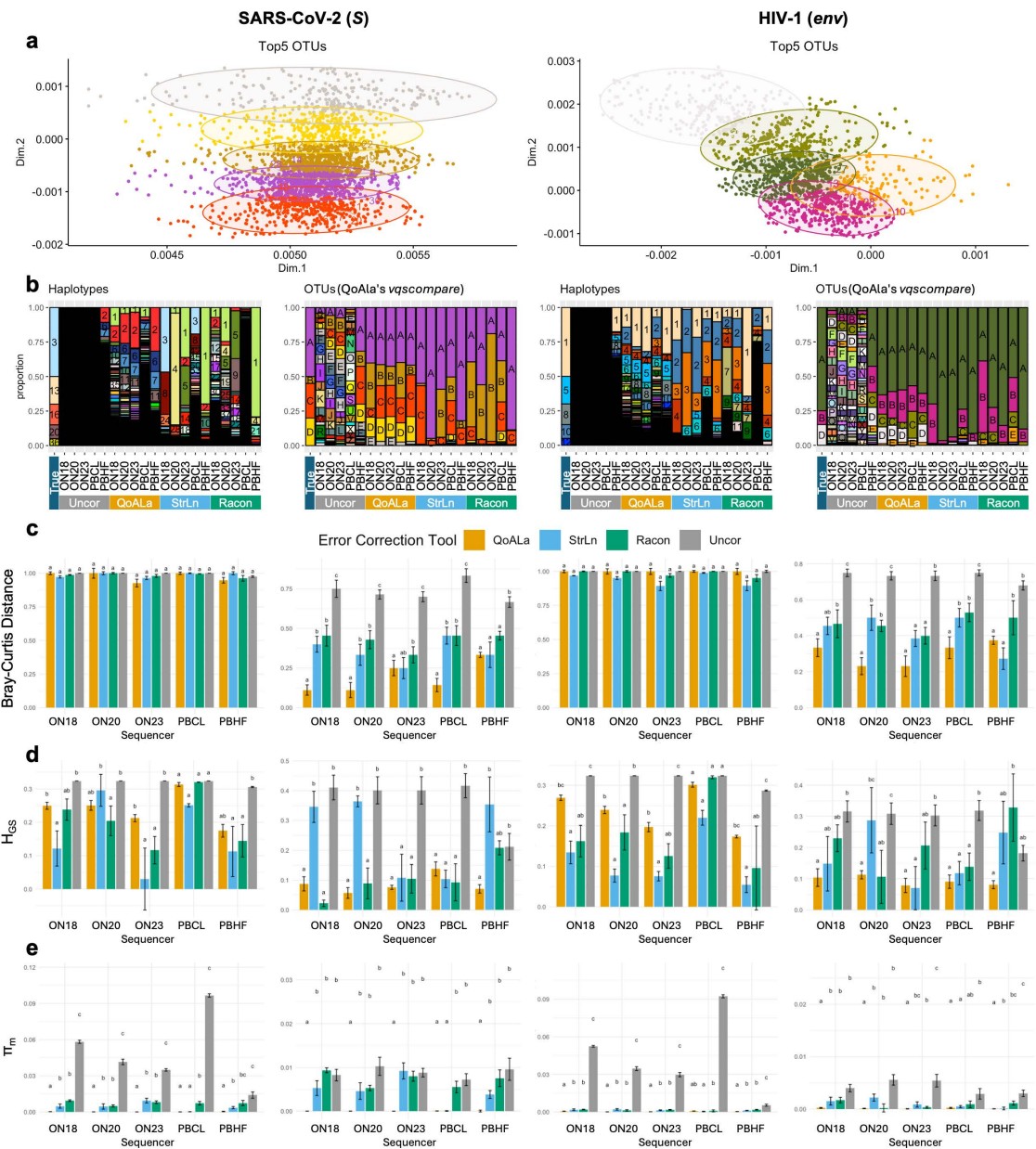

**Fig 2. Benchmarking performance on simulated datasets.** The left half of the figure shows results for SARS-CoV-2 (*S* gene) and the right half for HIV-1 (*env* gene); within each half, haplotype-level results are shown on the left and OTU-level results on the right. **a**, Multidimensional scaling (MDS) plots based on pairwise SNV distances among haplotypes (points), with ellipses indicating the five largest OTUs in one replicate. **b**, Stacked bar charts showing the relative abundance of haplotypes and OTUs, using the same color scheme as in panel a, across the ground truth, uncorrected reads, and reads corrected by different methods in the same replicate as a. **c**, Median Bray–Curtis dissimilarity, **d**, Median deviation in the Gini–Simpson diversity index ($H_{GS}$), and **e**, Median deviation in nucleotide diversity ($\pi_m$), each calculated across five replicates between correction methods and sequencing platforms relative to the ground truth. Letters in panels c–e denote statistically significant differences between methods (FDR < 0.05). Sequencing platform abbreviations are as follows: ON18, Oxford Nanopore Technologies 2018; ON20, Oxford Nanopore Technologies 2020; ON23, Oxford Nanopore Technologies 2023; PBCL, PacBio continuous long reads (CLR); PBHF, PacBio HiFi.

## Effect of down-sampling on diversity metrics

While read down-sampling is necessary for standardizing sample size before comparison, it may alter viral quasispecies diversity measures, especially at very low sample sizes. To assess this impact, we sub-sampled the real dataset reads with replacement and computed nine diversity metrics. Downsampling ranged from 10,000 down to 25 reads, depending on the original depth of each read alignment sample (Methods). This process was repeated 100 times per sub-sample size. This assessment was conducted twice: once for the alignments that were sampled after noise-minimization, and another for the alignments that were sampled before noise-minimization.

The Gini–Simpson index ($H_{GS}$) and nucleotide diversity ($\pi_m$) were the most robust diversity metrics, remaining relatively stable across all sample sizes regardless of the original alignment depth (Fig 3). In contrast, diversity metrics directly reliant on read depth [number of haplotypes (H) and mutation frequency at the molecular level (Mfm)], indirectly reliant on read depth [Shannon entropy ($H_S$), mutation frequency at the molecular level (Mfe), functional attribute diversity (FAD),

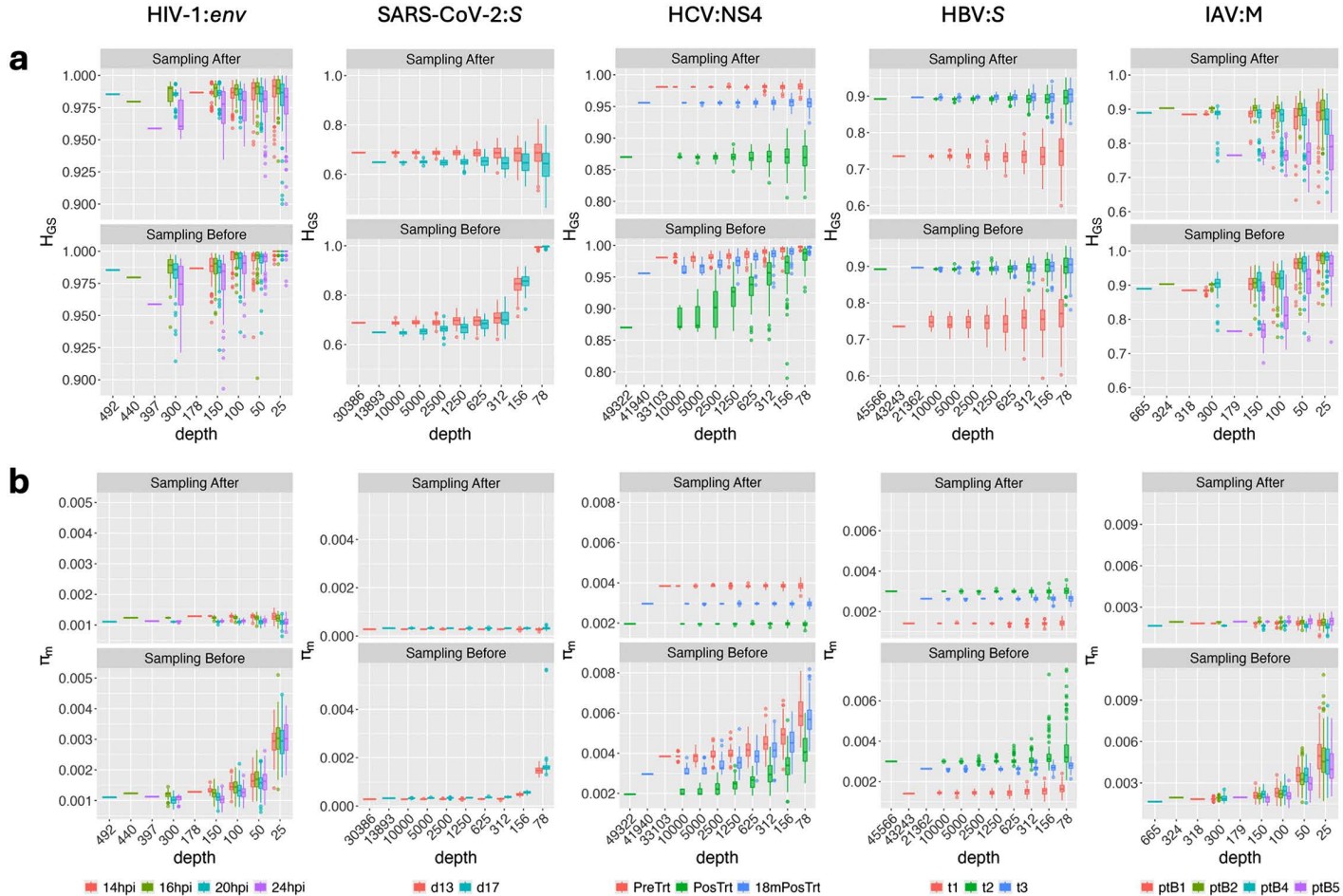

**Fig 3. Effect of down-sampling on key diversity metrics for five viruses (virus:gene). a,** Distribution of the Gini-Simpson index ($H_{GS}$) for each viral species: gene samples computed at different sample sizes (100 repeated random samplings with replacement for each size) after (top) and before (bottom) noise-minimization. **b,** Distribution of nucleotide diversity ($\pi_m$) with a layout similar to that of panel **a.** Bottom legends indicate the names of samples for each virus dataset. For all plots, horizontal lines indicate medians, boxes the 25th to 75th percentile, whiskers the lowest and highest values excluding outliers, and dots the outliers. Each dot indicates the mean of one target, boxes the 25th to 75th percentile, lines medians and whiskers extend from minimum to maximum values.

and nucleotide diversity at the entity level ($\pi_e$)], or normalized by read depth [normalized Shannon entropy ($H_{SN}$)] were impacted to varying degrees by sample size (S2–S11 Figs). Down-sampling post noise-minimization from high-depth alignments (>10,000 reads) minimally affected $H_S$ and Mfm at sample sizes >1,000 reads (S2, S4, and S8 Figs), whereas sampling from low-depth alignments (<1,000 reads) resulted in abrupt changes in these metrics at any sample size (S6 and S10 Figs). Computing metrics from alignments sampled after noise-minimization generally yielded less varied values compared to those sampled before noise-minimization in most instances (S3, S5, S7, S9, and S11 Figs).

Regarding inter-sample comparisons, variations in metrics due to sample size and sampling strategy marginally affected comparisons in scenarios where the actual diversity measures of samples were either highly similar (Figs 3 and S6–S11) or markedly different (Figs 3 and S2–S5). However, down-sampling to very low sample sizes (<100 reads) or down-sampling before noise-minimization likely resulted in false equivalences or differences because of the high variation in computed diversity metrics (Figs 3 and S2–S11). Thus, our analysis suggests that diversity metrics from samples with read depths of <100 should be interpreted with caution.

## Example scenarios of diversity comparison

Since a single mutation can differentiate one haplotype from another, and such mutations may introduce noise and obscure the true frequency of different major genetic variants within the quasispecies, solely using haplotype-based diversity metrics to summarize the viral quasispecies profile might hinder our ability to observe the continuous dynamics of the virus across samples. To illustrate this, we compared related viral samples in various scenarios using both traditional haplotyping based on a strict mutational profile and a novel OTU assignment based on genetic distance (Methods). In the latter, haplotypes are clustered by relatedness into larger operational taxonomic units (OTUs). This allows the user to better comprehend the structure of the quasispecies, i.e., the evolving genetic diversity surrounding several main variants within the viral cloud.

The studies on HIV-1 and SARS-CoV-2 served as examples of short-term longitudinal sampling scenarios. Within a few hours of each other, *env* genes of four HIV-1 samples infecting T cells in vitro exhibited numerous haplotypes, among which a common dominant haplotype was not found (Fig 4a). When OTU clustering was applied to the same samples, the members and proportions of all four samples were relatively similar (Fig 4b), with the 186 haplotypes being optimally clustered into 10 OTUs (Fig 4c and 4d). Examining five SARS-CoV-2's *S* gene samples taken within a nine-day period from a single patient, the dominant haplotype and OTU occupied over half of the virus population in each sample (Fig 4a and 4b), with minor haplotypes appearing as outliers (Fig 4c and 4d). It is also worth noting that the sporadic appearance of many minority variants in the day 13 and 17 SARS-CoV-2 samples (Fig 4a and 4b) was likely due to noise rising from incomparable read depths in the original alignments, which were over ten times lower than the other three samples (S5 Table).

As a model for long-term longitudinal sample comparison, studies on HCV and HBV were utilized. Analysis of the NS4 coding region of HCV revealed that viral haplotypes and their proportions underwent sudden changes after treatment. Interestingly, certain major haplotypes detected pre-treatment reappeared 18 months post-treatment (Fig 4a). Concurrently, OTU clustering provided better insights into the dynamics of HCV across three sampling points by grouping genetically closely related minor haplotypes into common clusters observed at varying proportions across all time points (Fig 4b, 4c and 4d). The within-host virus population dynamics were also clearly illustrated by HBV's *S* gene. Both haplotype and OTU classifications underscored the gradual turnover of predominant haplotypes from timepoint one to four over a 30-month period, and a completely different population emerged at timepoint five, occurring over 80 months after the initial sampling (Fig 4a and 4b).

Lastly, in the scenario of the outbreak represented by IAV's M gene segment samples, viruses were collected cross-sectionally among patients in two wards (B and C) within the same hospital. At the haplotype level, only two mostly identical sets of virus populations were found in more than one patient (B1, B2 and C1, C3), without other prominent links between cases or wards (Fig 4a). However, upon regrouping them into new OTUs, we discovered that all patients in ward B were partly infected with the same genetically similar sub-population, while all patients in ward C were infected with

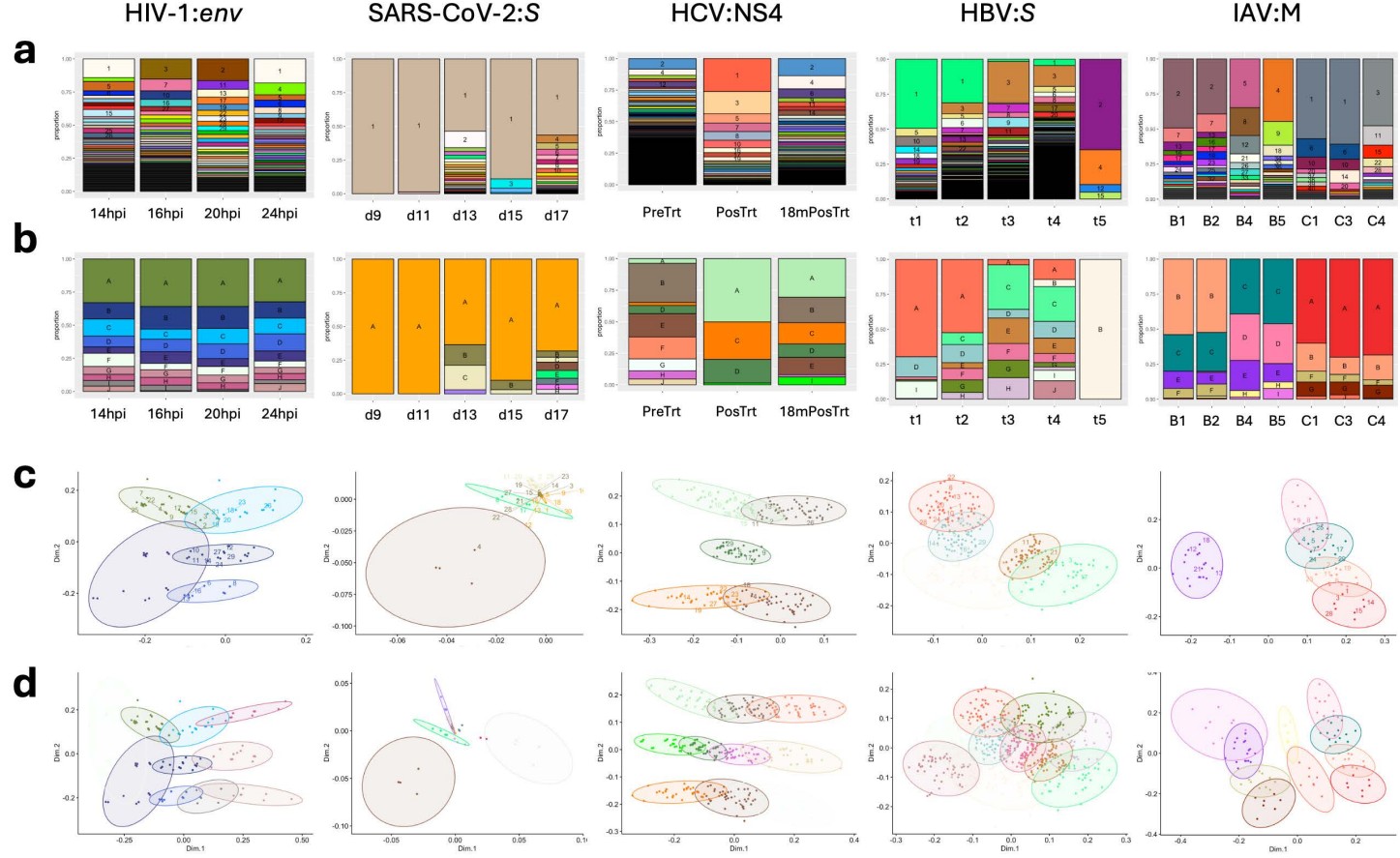

**Fig 4. Examples of viral quasispecies and OTU comparisons for five viruses (virus:gene). a,** Stacked bar chart presenting the proportion of each unique haplotype (same color) identified across different samples within each viral species: gene dataset. **b,** Stacked bar chart presenting the proportion of new OTUs, each consisting of genetically closely related haplotypes. **c,** Multidimensional scaling (MDS) plot of haplotypes' pairwise SNV distance (dot), showing the five largest OTUs (circle) with a color scheme corresponding to plot b. **d,** MDS plot of all OTUs.

a similar set of OTUs. Furthermore, the predominant OTU found in two ward B patients (B1, B2) was also present as a minority group in all ward C patients, suggesting potential inter-ward transmission of IAV (Fig 4b, 4c and 4d).

The comparative profiles of viral quasispecies, whether analyzed at the haplotype or OTU levels, based on other genes from the same viral samples (including HBV's *P* gene, HIV-1's *gag* gene, and SARS-CoV-2's *ORF3a* gene), exhibited visual similarities to the chosen genes depicted in Fig 4 (S12-S14 Figs). This suggests the absence of linkage disequilibrium between the observed genes of these three viruses, indicating that they may adequately represent the population dynamics at the whole-genome level. Interestingly, one comparable result between the original and our studies is the number and frequency of haplotypes found in the day 11 and day 15 SARS-CoV-2 samples. We captured a similar quasispecies diversity in the *S* gene samples and discovered two SNV positions in the *ORF3a* gene that created two rare haplotypes not reported in the original study (S7 Table and Ko et al., 2021 [45]).

## Discussion

The *longreadvqs* package was developed as a tool for both quantitative and qualitative analysis of viral quasispecies diversity, addressing the challenges posed by varied error rates and read depths inherent in long-read sequencing

technologies as well as noise introduced by rare SNVs in the alignment. During the testing phase, we carefully considered these variations across diverse sampling scenarios and multiple viral species, ensuring the tool's general applicability. The strengths of this tool lie in its ability to customize parameter settings based on prior information and in providing comprehensive visualizations that elucidate the dynamics of virus microevolution across multiple samples.

The analytical findings from real datasets derived from the QoALa workflow presented in this study provide unique perspectives and insights not previously explored in the original research. For instance, we utilized raw read data to reconstruct within-host virus haplotypes and OTUs, emphasizing an overview of population structure changes over time or among outbreak cases. In contrast, the original studies focused on specific aspects, such as mutation spectra [46], deletions [47], gene splicing [48], or phylogenetic analysis of consensus sequences [49]. To build read alignments, we employed a uniform genome assembly workflow for all raw read samples, which was not tailored or reproduced from the original protocols created for each dataset. Furthermore, in the comparison step, we fixed the number of sample sizes and k-means clusters for all scenarios. Hence, discrepancies in details between the original findings and ours may have arisen, which we acknowledge as a limitation on our part. Nevertheless, it's essential to note that the objective of our study does not revolve around delivering new findings from each dataset but rather demonstrating the usage and versatility of this package.

Previous studies have established a robust foundation in measuring viral quasispecies diversity and developing analytical workflows for NGS data over the past decade [22,42,43,50]. However, much of this research cannot be directly applied to long-read sequencing data. Long-read quasispecies analyses are hindered by the significantly higher level of noise present in long-read data driven by the higher likelihood that longer reads will have at least one SNV across their length. For NGS short-read data, it is recommended to employ techniques such as rarefaction, resampling, and fringe trimming based on haplotype frequency to estimate appropriate sample sizes and minimize bias from unbalanced samples before comparing diversity metrics [43]. The presence of a misrepresented large proportion of singleton haplotypes (S1 Fig), compounded by the high sequencing error rate of current long-read technologies, hinders us from following such approaches.

To systematically evaluate the QoALa workflow against existing error-correction approaches for long read data, we benchmarked its performance on simulated SARS-CoV-2 and HIV-1 quasispecies datasets across multiple sequencing platforms. QoALa consistently outperformed alternative methods in recovering accurate quasispecies composition at the OTU level across Bray-Curtis dissimilarity, Gini-Simpson index, and nucleotide diversity metrics. This advantage likely stems from fundamental differences in approach: Strainline is a *de novo* assembler designed to reconstruct strain-resolved, full-length viral haplotypes [33], while Racon is a standalone consensus module intended to polish raw contigs generated by rapid assembly methods using partial order alignment [44]. Neither tool explicitly addresses the characteristic accumulation of singleton haplotypes in viral quasispecies data from noisy long reads. QoALa instead optimizes its noise-minimization cut-off based on singleton haplotype frequency—targeting the specific aspect caused by error signature in long-read viral sequencing. While QoALa showed reduced performance at the individual haplotype level compared to Strainline and Racon, this trade-off is justified for viral quasispecies analysis where biologically meaningful diversity is better captured through OTU-level clustering that groups genetically related variants—an approach we demonstrate throughout this study reduces the impact of residual sequencing noise while preserving evolutionarily relevant population structure. Importantly, the value of the QoALa workflow extends beyond its current noise-minimization strategy. While existing tools provide only corrected reads or reconstructed haplotypes without integrated downstream analytical capabilities, QoALa offers comprehensive functionality for cross-sample comparison, OTU-based regrouping of genetic variants, and interactive visualizations that illuminate quasispecies population dynamics. These analytical and visualization features remain essential for interpreting viral diversity regardless of sequencing technology improvements or the emergence of superior error-correction methods, ensuring the workflow's continued relevance even as the field advances toward platforms that may eventually render the noise-minimization step unnecessary.

Building on these comparative advantages, we designed QoALa's noise-minimization strategy to address the specific challenges of long-read viral quasispecies data. Instead of excluding low-frequency haplotypes as fringe trimming does, we opt to retain all reads and replace low-frequency SNVs with the majority base at their respective positions. This approach offers two primary benefits. Firstly, low-frequency haplotypes, comprising both true mutations and errors, are not eliminated from the alignment; rather, potential erroneous SNVs within the haplotype are smoothed to the mode. This advantage is particularly crucial when comparing shallow depth read alignments, where removing reads belonging to low-frequency haplotypes would severely reduce sample size. Secondly, the false haplotype diversity resulting from long-read errors are mitigated by consolidating previously low-frequency haplotypes into larger haplotype groups (OTUs) that share common SNVs, which are more likely to represent true mutations and provide more clarity on the structured diversity existing within quasispecies.

However, it is worth noting that our position-wise noise minimization approach carries inherent limitations in co-infection scenarios. Because corrections are determined by per-position nucleotide frequencies across the entire read pool, a genuine minority-strain variant may be incorrectly corrected toward the majority-strain consensus, or a sequencing error may escape correction if it coincidentally matches a co-infecting strain's variant frequency. This limitation is less pronounced under "*conbase*" (the default method)—which replaces low-frequency bases with the position-wise majority base—than under "*domhapbase*", which anchors all corrections to the single dominant haplotype and would more aggressively pull corrected reads toward the dominant strain at the expense of the minority. Both methods are nevertheless considerably more conservative than whole-haplotype replacement or removal approaches, which would eliminate minority haplotypes entirely. Consistent with this design, our benchmarking on simulated datasets showed that the approach preserved nucleotide diversity and the hierarchical population structure of quasispecies more accurately than alternative error-correction and haplotype reconstruction methods.

Many factors need to be considered in selecting the cut-off percentage for position-wise noise-minimization. Factors such as gene mutation rate, sequencing error rate, and the trade-off between retaining versus unifying rare haplotypes are critical. Especially when dealing with a lengthy alignment where singleton haplotypes are more common, one may have to decide whether to prioritize retaining most mutation details, thus making it difficult to quantify overall population complexity, or to focus on the broader picture of viral cloud diversity while potentially discarding some details that may or may not be significant.

Standardizing unbalanced sample sizes by down-sampling can introduce bias in diversity comparisons, especially when integrating noise-minimization. Since sampling techniques used for NGS data [43] are not appropriate for long-read data, we redesigned sampling strategies by integrating the concept of rarefaction and repeat sampling, taking into account the order of sampling before or after noise-minimization to simulate possible options for analysis. Here, we demonstrate the sensitivity of diversity metrics across our example scenarios. Based on this analysis, down-sampling after noise-minimization is our recommended technique. However, repeated down-sampling to observe the distribution of metrics at different sample sizes, at least between the largest and smallest samples in the dataset, is encouraged to better understand the sensitivities of diversity metrics to sampling depth in a particular project. Ultimately, we suggest interpreting quantitative metrics comparisons alongside qualitative profiles that uncover shared viral variants between related samples, either at the haplotype or OTU levels, to better illuminate the dynamics of viral quasispecies, as made possible by the QoALa workflow implemented in the *longreadvqs* package.

## Methods

### Simulated sequencing data

To benchmark our analytical pipeline under controlled conditions, we generated simulated viral long-read sequencing datasets for severe acute respiratory syndrome coronavirus 2 (SARS-CoV-2) and human immunodeficiency virus (HIV-1). For each virus, we created five independent replicates, each containing five haplotypes with fixed frequency distributions

of 50%, 20%, 15%, 10%, and 5%. Mutations were introduced into reference genomes (NCBI accessions listed in S1 Table at virus-specific rates. For SARS-CoV-2, we applied a single nucleotide polymorphism (SNP) rate of $8 \times 10^{-4}$ and indel rate of $5 \times 10^{-5}$ per site [51]. For HIV-1, we used a SNP rate of $2 \times 10^{-3}$ and indel rate of $3 \times 10^{-4}$ per site [52]. Each replicate was generated with a unique random seed to produce distinct mutation patterns while maintaining consistent haplotype proportions across replicates. For each replicate, we first generated error-free "true reads" at $500 \times$ coverage to serve as ground truth. We then simulated platform-specific sequencing reads at $10 \times$ coverage using Badread v0.4.1 [53], modeling five sequencing technologies with their characteristic error profiles: Oxford Nanopore Technologies (ONT) 2018, 2020, and 2023 models, and PacBio continuous long reads (CLR) and highly accurate long sequencing reads (HiFi). The simulated dataset comprised 50 sets of reads (2 viruses × 5 replicates × 5 sequencing platforms).

### Real sequencing data

To evaluate our package and workflow, we gathered raw sequencing reads from five projects focusing on whole genome sequence (WGS) of five different viruses: SARS-CoV-2, HIV-1, hepatitis C virus (HCV), hepatitis B virus (HBV), and influenza A virus (IAV), which are among the most studied species in the field of viral quasispecies. These reads were generated using either PacBio circular consensus sequencing (CCS) or ONT and were sourced from the National Center for Biotechnology Information's Sequence Read Archive (NCBI SRA). Specifically, we selected projects that obtained either longitudinal viral samples or cross-sectional samples from a single outbreak, where the genetic relationships were suitable for comparison at the quasispecies level. PacBio read samples were obtained from three projects: three HCV samples from a chronic hepatitis patient collected before and after treatment over 18 months [46]; five HBV samples collected at various time points within 100 months from an untreated patient [47]; and five longitudinal SARS-CoV-2 samples collected between 8–17 days after clinical onset from a single patient [45]. ONT data included four HIV-1-infected T cell samples collected up to 24 hours post-infection [48] and seven IAV samples from different patients during a nosocomial outbreak [49]. The NCBI accession numbers of all 24 raw read samples across five projects are listed in S1 Table.

Raw reads were processed using comparable workflows for both sequencing platforms. Adapter sequences were trimmed using HiFiAdapterFilt v3.0.1 [54] for PacBio reads and Porechop v0.2.4 [55] for ONT reads. Quality filtering was performed using Filtlong v0.2.1 [56] to remove PacBio reads shorter than 1,000 bp or ranking in the lowest 10% by composite read score (a weighted combination of read length and mean quality); since all reads across datasets had mean quality scores at or above Q30, the relative filter removed reads based on length rather than quality. NanoFilt v2.8.0 [57] was used to remove ONT reads with average quality scores < 7. Trimmed and filtered reads were aligned to NCBI reference genomic sequences (S1 Table) using minimap2 v2.26 [58] with technology-appropriate presets ("map-hifi" for PacBio CCS and "map-ont" for ONT). SAM files from read mapping were converted to FASTA format and fragmented into genes or coding regions based on genomic annotations using SeqKit v2.7.0 [59], retaining only reads covering the full length of each gene or region (Fig 1a). The depth and length of final alignments are documented in S2-S6 Tables.

### The *longreadvqs* package and QoALa workflow

The *longreadvqs* package implements the QoALa workflow, a standardized framework for generating noise-minimized alignments, harmonizing sequencing depth, extracting haplotypes and operational taxonomic units (OTUs), and performing cross-sample viral quasispecies comparisons. The workflow begins with importing equal-length read alignments from any genome assembly or long-read processing pipeline. Noise minimization is performed using the "*vqsassess*" function, which replaces low-frequency single-nucleotide variants (SNVs) with either the majority nucleotide ("*conbase*" method) or, when applicable, the nucleotide from the dominant haplotype ("*domhapbase*" method) (Fig 1b). The frequency threshold can be set based on prior knowledge, such as the estimated sequencing error rate or virus-specific mutation rates, or optimized empirically using the "*pctopt*" function. "*pctopt*" shows the percentage of singleton haplotypes (haplotypes represented by a single read) across candidate thresholds, allowing identification of the optimal cut-off where this percentage

plateaus (elbow point) (Fig 1a). When analyzing multiple samples, "*vqsassess*" also performs random downsampling to the lowest sequencing depth to reduce bias in diversity comparisons.

Haplotypes are defined as unique nucleotide sequences across the full-length alignment of the analyzed gene or genomic region. After noise minimization, reads with identical sequences are grouped into the same haplotype, and haplotype frequencies are calculated as the proportion of reads assigned to each haplotype relative to the total number of retained reads. These haplotypes and their frequencies are extracted for each sample and evaluated using the "*snvcompare*" function. For each sample, SNV frequencies across the alignment are summarized from the read data produced by "*vqsassess*", and the resulting SNV frequency profiles are plotted and arranged across samples to enable visual comparison of variant patterns and positional consistency along the sequence (Fig 1c). Because sequencing noise can vary along the gene or genomic region—for example, in homopolymer regions [60,61] or at soft-clipped read ends [62,63]—the "*vqscustompct*" function allows region-specific adjustment of the noise-minimization cut-off to improve accuracy (Fig 1b).

Processed samples are then integrated using "*vqscompare*" (Fig 1d), which identifies haplotypes shared across samples and visualizes unique haplotypes per sample in color-coded bar plots. Haplotype sequences are reclassified into OTUs based on genetic distances calculated from the SNV alignment using dist.dna (ape v5.7.1 [64]). The distance matrix is converted into dissimilarity coordinates and clustered using classical multidimensional scaling (cmdscale) followed by k-means clustering (kmeans) in stats v4.3.1 [65]. The number of clusters is user-defined, and OTU composition is visualized in bar plots, with MDS plots illustrating clusters and major haplotypes within each OTU. All resulting plots are generated using ggplot2 v3.4.4 [66].

In addition to visualizations, "*vqscompare*" outputs noise-minimized, downsampled read and SNV alignments, classified haplotypes and OTUs, and quasispecies diversity metrics. Nine viral quasispecies diversity metrics are calculated per sample using QSutils v1.18.0 [67], with an additional set based on OTU consensus reads and frequencies for large-scale diversity assessment. Together, these functions constitute the QoALa workflow, providing a reproducible, standardized approach for viral quasispecies characterization, sample harmonization, and cross-sample comparison.

## Benchmarking analysis using simulated sequencing data

For benchmarking, the performance of the QoALa workflow was compared against three alternative approaches: Strainline-corrected reads, Racon-corrected reads, and uncorrected raw reads. Strainline represents a platform specifically designed to correct and reconstruct haplotypes from viral quasispecies [33], whereas Racon is a general-purpose long-read error-correction tool [44]. Both tools are applicable to ONT and PacBio sequencing data. All 50 simulated read sets for each virus were corrected using either Strainline (with sequencing-platform-specific parameters) or Racon v1.5.0 (with default settings). All uncorrected (n = 50) and corrected (n = 100) simulated datasets were aligned to their respective reference viral genomes using minimap2 v2.26 [58] with technology-appropriate presets ("map-ont" for ONT; "map-pb" or "map-hifi" for PacBio). SAM files were converted to FASTA format, and gene-specific regions were extracted using SeqKit v2.7.0[59]—the spike (*S*) gene for SARS-CoV-2 and the envelope (*env*) gene for HIV-1. Only reads spanning the full gene length were retained for downstream analyses.

For the aligned uncorrected S and env reads, noise-minimization thresholds for the QoALa workflow were estimated with "*pctopt*", selecting the median percentage at which singleton haplotype proportions plateaued across replicates for each platform—HIV-1 *env*: ONT 2018 (12%), ONT 2020 (5%), ONT 2023 (6%), PacBio CLR (9%), PacBio HiFi (1%); SARS-CoV-2 *S*: ONT 2018 (13%), ONT 2020 (6%), ONT 2023 (8%), PacBio CLR (8%), PacBio HiFi (1%) (S15 Fig). These thresholds were then passed to "*vqsassess*", which imported all corrected and uncorrected read sets and uniformly downsampled them to a depth of 500 to match the error-free ground-truth coverage. Noise minimization was applied only to the uncorrected reads designated for QoALa's threshold-based correction, while corrected reads and comparison uncorrected sets were downsampled but left otherwise unmodified.

A total of 200 aligned read sets (2 viral genes × 5 replicates × 5 sequencing platforms × 4 correction approaches) were subsequently compared with the two corresponding ground-truth viral quasispecies (100 aligned reads per ground truth) using "*vqscompare*". Comparisons were conducted at both the haplotype and OTU levels. For OTU-level analyses, haplotypes were reclustered into 30 OTUs per gene, and resulting OTU frequency profiles were used as the basis for evaluation.

Quasispecies reconstruction accuracy was evaluated using three metrics. Bray-Curtis dissimilarity [68] was calculated between each reconstructed sample and its corresponding ground truth using "*vegdist*" in *vegan* R package v2.6.8 [69] to provide a direct comparative measure incorporating both qualitative similarity (presence of shared haplotypes or OTUs) and quantitative agreement (their relative abundances); Bray-Curtis ranges from 0 (identical composition) to 1 (completely dissimilar), with lower values indicating more accurate reconstruction. Within-sample diversity was summarized using the Gini-Simpson index ($H_{GS}$), where larger values indicate greater diversity and evenness; deviations from the ground-truth $H_{GS}$ therefore reflect structural differences rather than specific compositional mismatches. Nucleotide diversity ($\pi_m$) was also computed as a sequence-based measure of genetic/functional diversity, where higher values reflect more divergent viral populations; although $\pi_m$ is not a direct pairwise-comparative index, its deviation from the true $\pi_m$ indicates how well each method captures underlying heterogeneity.

Statistical comparisons were conducted on per-replicate deviations from the ground truth for each metric across correction methods (QoALa, Strainline, Racon, and uncorrected reads) and across sequencing technologies. Two-sided Wilcoxon rank-sum tests (wilcox.test) [65] were applied independently for each metric and for both haplotype- and OTU-level analyses to accommodate small replicate sizes and non-normal deviation distributions. To improve reproducibility, p-values were adjusted using the Benjamini-Hochberg false discovery rate (FDR) procedure via p.adjust [65]; significance was defined at FDR < 0.05. Because Bray-Curtis dissimilarity decreases with accuracy, and $H_{GS}$ and $\pi_m$ increase with diversity, smaller deviation values indicate better agreement with the ground-truth population structure. This statistical framework identified correction methods that yielded significantly smaller deviations—and therefore more accurate reconstructions—than uncorrected reads and other correction approaches across sequencing platforms. All visualizations of metric comparisons and statistical outcomes were generated using ggplot2 v3.4.4 [66].

## Down-sampling sensitivity analysis and viral quasispecies comparison for real sequencing data

To assess how sequencing depth influences diversity estimation and quasispecies comparison in the QoALa workflow, we conducted a sensitivity analysis using real sequencing datasets. We extracted read alignments from specific genes or regions of five representative viral species. Alignments were selected based on having lengths exceeding or closely approximating 1,000 bases, coverage depths greater than 100 reads, and soft-clipped regions—when present—constituting less than 25% of the total alignment length. Using "*pctopt*", we determined the optimal noise-minimization cut-off percentage as previously described (Fig 1a). The selected alignments and their corresponding cut-off thresholds were: NS4 (15%) for HCV; *P* (5%) and *S* (5%) for HBV; *env* (15%) and *gag* (19%) for HIV-1; *S* (1%) and *ORF3a* (1%) for SARS-CoV-2; and the M segment (10%) for IAV (S1 Fig).

For high-depth samples (HCV, HBV, SARS-CoV-2), down-sampling series through "*vqsassess*" began at 10,000 reads and were iteratively halved until reaching a minimum of 78 reads. For low-depth samples (HIV-1, IAV), fixed subsample sizes of 300, 150, 100, 50, and 25 reads were used. For each down-sample size, random resampling with replacement was performed 100 times after noise minimization; an identical repeated-sampling protocol was also applied before noise minimization to directly compare the impact of noise reduction on depth sensitivity. For every iteration, nine diversity metrics from QSutils—fully described by Gregori et al.[42]—were computed: number of haplotypes (H), Shannon entropy ($H_S$), normalized Shannon entropy ($H_{SN}$), Gini-Simpson index ($H_{GS}$), functional attribute diversity (FAD), mutation frequency at the entity level (Mfe), nucleotide diversity at the entity level ($\pi_e$), mutation frequency at the molecular level (Mfm), and nucleotide diversity ($\pi_m$). Resulting metric distributions were visually compared across subsample sizes and before/after noise minimization to evaluate metric stability and the degree of information loss due to aggressive down-sampling.

Apart from the sensitivity analysis, we performed viral quasispecies comparisons among within-species/gene samples to demonstrate the application of the QoALa workflow to real experimental and clinical data. Noise-minimized, down-sampled alignments for each selected gene and viral species—using the pctopt-optimized thresholds described above and down-sampled to either the lowest observed depth for that gene or to 1,000 reads if the lowest depth exceeded 1,000 (S2–S6 Tables)—were generated using the "*vqsassess*" function and used as inputs for individual "*vqscompare*" analyses. The number of clusters (k) for OTU classification via k-means clustering was fixed at 10 for all runs. The resulting summary plots illustrate viral quasispecies dynamics across a range of biological scenarios, including short-term (hours to days in HIV-1 and SARS-CoV-2 datasets), long-term (months to years in HCV and HBV datasets) longitudinal sampling, and samples collected from the same outbreak cohort (IAV dataset).

## Supporting information

**S1 Fig. Selection of noise-minimization cut-off percentage based on percentage of singleton haplotypes.** Connected scatter plot with points indicating the percentage of singleton haplotypes (y-axis: pctsingleton) at different noise-minimization cut-off percentages (x-axis: pct). Lines and dots are colored by individual samples. Vertical dashed lines represent selected cut-off percentages for each gene and viral species. The high proportion of singleton haplotypes observed in some datasets (e.g., HCV NS4), despite being generated using high-accuracy PacBio CCS sequencing, likely reflects the combined effects of residual sequencing errors at the haplotype level and genuine within-host viral diversity, particularly in chronic infections such as HCV where even single-nucleotide differences can define distinct haplotypes. (DOCX)

**S2 Fig. Effect of down-sampling after noise-minimization on nine diversity metrics in HCV NS4 samples.** Distribution of number of haplotypes (H), Shannon entropy ($H_s$), normalized Shannon entropy ($H_{SN}$), Gini-Simpson index ($H_{GS}$), functional attribute diversity (FAD), mutation frequency at the entity level (Mfe), nucleotide diversity at the entity level ($\pi_e$), mutation frequency at the molecular level (Mfm), and nucleotide diversity ($\pi_m$) computed at different sample sizes (100 repeated random samplings with replacement). Horizontal lines indicate medians, boxes 25th–75th percentile, whiskers min–max (excluding outliers), dots outliers. (DOCX)

**S3 Fig. Effect of down-sampling before noise-minimization on nine diversity metrics in HCV NS4 samples.** Same layout as S2 Fig. (DOCX)

**S4 Fig. Effect of down-sampling after noise-minimization on nine diversity metrics in HBV *S* gene samples.** Same layout as S2 Fig. (DOCX)

**S5 Fig. Effect of down-sampling before noise-minimization on nine diversity metrics in HBV *S* gene samples.** Same layout as S2 Fig. (DOCX)

**S6 Fig. Effect of down-sampling after noise-minimization on nine diversity metrics in HIV-1 *env* samples.** Same layout as S2 Fig. (DOCX)

**S7 Fig. Effect of down-sampling before noise-minimization on nine diversity metrics in HIV-1 *env* samples.** Same layout as S2 Fig. (DOCX)

**S8 Fig. Effect of down-sampling after noise-minimization on nine diversity metrics in SARS-CoV-2 *S* gene samples.** Same layout as S2 Fig.
(DOCX)

**S9 Fig. Effect of down-sampling before noise-minimization on nine diversity metrics in SARS-CoV-2 *S* gene samples.** Same layout as S2 Fig.
(DOCX)

**S10 Fig. Effect of down-sampling after noise-minimization on nine diversity metrics in IAV M gene samples.** Same layout as S2 Fig.
(DOCX)

**S11 Fig. Effect of down-sampling before noise-minimization on nine diversity metrics in IAV M gene samples.** Same layout as S2 Fig.
(DOCX)

**S12 Fig. Viral quasispecies and OTU comparisons among HBV *P* and *S* gene samples.** From top to bottom: stacked bar chart of unique haplotype proportions; stacked bar chart of new OTUs; MDS plot of haplotypes' pairwise SNV distance highlighting five largest OTUs; MDS plot of all OTUs.
(DOCX)

**S13 Fig. Viral quasispecies and OTU comparisons among HIV-1 *env* and *gag* gene samples.** Same layout as S12 Fig.
(DOCX)

**S14 Fig. Viral quasispecies and OTU comparisons among SARS-CoV-2 *S* and *ORF3a* gene samples.** Same layout as S12 Fig.
(DOCX)

**S15 Fig. Selection of noise-minimization cut-off percentage based on singleton haplotype frequency in simulated data.** Connected scatter plots showing percentage of singleton haplotypes (y-axis) versus noise-minimization cut-off (x-axis). Lines and points colored by sample; vertical dashed lines indicate median cut-off for each sequencing platform and virus:gene combination.
(DOCX)

**S1 Table. List of SRA raw read samples from five different projects/viral species used for testing the longreadvqs package.**
(XLSX)

**S2 Table. Depth of coverage by coding regions of HCV samples.**
(XLSX)

**S3 Table. Depth of coverage by genes of HBV samples.**
(XLSX)

**S4 Table. Depth of coverage by genes of HIV-1 samples.**
(XLSX)

**S5 Table. Depth of coverage by genes of SARS-CoV-2 samples.**
(XLSX)

**S6 Table. Depth of coverage by gene segments of IAV samples.**
(XLSX)

**S7 Table. Single nucleotide variants (SNVs) and haplotypes found in the day 11 and 15 SARS-CoV-2 samples.**
(XLSX)

## Acknowledgments

We thank J. Gregori and M. Guerrero-Murillo from VHIR Vall d'Hebron Research Institute for their support in trouble-shooting the dependency of our package. We thank TGS. Williams from Guy's and St Thomas' NHS Foundation Trust for providing additional information of the IAV study. Individually, we thank D. Makau and J. Baker for testing the package.

## Author contributions

**Conceptualization:** Nakarin Pamornchainavakul, Kimberly VanderWaal.

**Data curation:** Nakarin Pamornchainavakul.

**Formal analysis:** Nakarin Pamornchainavakul.

**Funding acquisition:** Kimberly VanderWaal.

**Methodology:** Nakarin Pamornchainavakul, Kimberly VanderWaal.

**Resources:** Kimberly VanderWaal.

**Software:** Nakarin Pamornchainavakul.

**Supervision:** Declan C Schroeder, Kimberly VanderWaal.

**Validation:** Nakarin Pamornchainavakul, Kimberly VanderWaal.

**Visualization:** Nakarin Pamornchainavakul.

**Writing – original draft:** Nakarin Pamornchainavakul.

**Writing – review & editing:** Nakarin Pamornchainavakul, Declan C Schroeder, Kimberly VanderWaal.

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
